# Inhibition of BMI-1 Induces Apoptosis through Downregulation of DUB3-Mediated Mcl-1 Stabilization

**DOI:** 10.3390/ijms221810107

**Published:** 2021-09-18

**Authors:** Kaixin Wu, Seon-Min Woo, Seung-Un Seo, Taeg-Kyu Kwon

**Affiliations:** 1Department of Immunology, School of Medicine, Keimyung University, Daegu 42601, Korea; 18242053482@163.com (K.W.); woosm724@gmail.com (S.-M.W.); sbr2010@hanmail.net (S.-U.S.); 2Center for Forensic Pharmaceutical Science, Keimyung University, Daegu 42601, Korea

**Keywords:** cancer stem-like cell, BMI-1, PTC596, Mcl-1, apoptosis

## Abstract

BMI-1, a polycomb ring finger oncogene, is highly expressed in multiple cancer cells and is involved in cancer cell proliferation, invasion, and apoptosis. BMI-1 represents a cancer stemness marker that is associated with the regulation of stem cell self-renewal. In this study, pharmacological inhibition (PTC596) or knockdown (siRNA) of BMI-1 reduced cancer stem-like cells and enhanced cancer cell death. Mechanistically, the inhibition of BMI-1 induced the downregulation of Mcl-1 protein, but not Mcl-1 mRNA. PTC596 downregulated Mcl-1 protein expression at the post-translational level through the proteasome-ubiquitin system. PTC596 and BMI-1 siRNA induced downregulation of DUB3 deubiquitinase, which was strongly linked to Mcl-1 destabilization. Furthermore, overexpression of Mcl-1 or DUB3 inhibited apoptosis by PTC596. Taken together, our findings reveal that the inhibition of BMI-1 induces Mcl-1 destabilization through downregulation of DUB3, resulting in the induction of cancer cell death.

## 1. Introduction

Cancer stem-like cells (CSCs), a small subpopulation of tumor cells, can initiate tumorigenesis through self-renewal and differentiation [1]. The tumor-initiating capacity of CSCs contributes to tumor relapse and resistance to chemotherapy and radiotherapy, thereby resulting in poor prognosis in cancer patients [2,3]. Therefore, dysregulated CSCs are significant candidates for successful cancer therapy. Recent studies have demonstrated that BMI-1 plays a role as a cancer stemness marker and is involved in the functional regulation of CSCs.

BMI-1 was initially identified as a transcription repressor belonging to the polycomb family [4]. Subsequently, many researchers have demonstrated that BMI-1 can have a major role in the self-renewal and differentiation of CSCs, including hematopoietic, leukemic, glioblastoma, and prostate cells [5,6,7,8]. Deletion of BMI-1 decreases sphere formation, whereas overexpression of BMI-1 increases sphere formation [8,9]. BMI-1 is involved in the plasticity, proliferation, and growth of CSCs through the regulation of the Ink4a/Arf, Notch, and Wnt/β-catenin signaling pathways [9,10]. Moreover, BMI-1 is upregulated in many cancers and is associated with proliferation, invasion, metastasis, apoptosis, and malignant transformation in cancer cells [11,12,13,14]. Therefore, BMI-1 as a significant prognostic marker may be an effective target for cancer therapy.

Many small-molecule BMI-1 inhibitors, such as PTC-209, PTC-028, and PTC596, suppress self-renewal ability, tumor growth, and cancer invasion [15]. In addition, these inhibitors induce cell death in various cancer cells [16,17,18,19]. PTC596 is a second-generation BMI-1 specific inhibitor with cell-permeable activity. Previous studies have reported that PTC596 increases mitochondrial apoptosis through the reduction of Mcl-1 expression in acute myeloid leukemia progenitor cells and mantle cell lymphoma [20,21]. Furthermore, PTC596 enhances sensitivity to various inhibitors, including Mcl-1, MEK, and tyrosine receptor inhibitors through Mcl-1 downregulation [19,21,22]. Although Mcl-1 downregulation by BMI-1 inhibition has been reported, the molecular mechanisms underlying Mcl-1 downregulation have not been completely investigated.

In this study, we investigated whether inhibition of BMI-1 using an inhibitor (PTC596) and siRNA induced apoptotic cell death through Mcl-1 destabilization caused by the decreased expression of DUB3.

## 2. Results

### 2.1. BMI-1 Inhibitor PTC596 Decreases the Cancer Stem-Like Cells (CSCs) Population

BMI-1 is associated with self-renewal of CSCs [6]. To determine whether the inhibition of BMI-1 can reduce the CSCs population, we examined the effect of the novel pharmacological BMI-1 inhibitor, PTC596, and BMI-1 siRNA. As expected, PTC596 and knockdown of BMI-1 decreased the CSC population in human renal (Caki) and cervical (HeLa) carcinoma (Figure 1A,B). In addition, we measured the levels of CD44 cancer stem cell markers using flow cytometry. PTC596 decreased the number of CD44-positive cells (Figure 1C). These results showed that inhibition of BMI-1 induces a decrease in the CSCs population.

### 2.2. PTC596 Induces Apoptosis through Caspase-3 Activation

Previous studies have reported that various BMI-1 inhibitors induce cancer cell death [17,18,19]. We examined the anti-cancer effects of two BMI-1 inhibitors (PTC596 and PTC209) in multiple cancer cells. Interestingly, PTC596 effectively induced death rather than PTC209 in Caki, HeLa, and human lung carcinoma (A549) cells (Figure 2A). In addition, PTC596 increased annexin V-positive cells (Figure 2B). Therefore, we investigated the effects of apoptosis and the molecular mechanisms of PTC596 in renal cancer cells. As shown in Figure 2C–E, PTC596 treatment showed typical apoptotic morphology, chromatin condensation, increase of DNA fragmentation, and activation of caspase-3 (DEVDase) activity. PTC596 increases DEVDase (caspase-3) activity. To verify the caspase-3-dependent apoptosis induced by PTC596, we pretreated the pan-caspase inhibitor, z-VAD-fmk (z-VAD). z-VAD prevented the apoptosis and cleavage of PARP as well as caspase-3 activation in PTC596-treated cells (Figure 2F). These results suggest that PTC596 induces caspase-dependent apoptosis in cancer cells.

### 2.3. PTC596 Downregulates Mcl-1 Expression at the Post-Translational Level

Next, we investigated the expression levels of apoptosis-related proteins using PTC596 and BMI-1 siRNA. PTC596 and BMI-1 siRNA only decreased Mcl-1 protein levels, whereas other proteins did not change (Figure 3A). However, Mcl-1 mRNA expression was not altered by BMI-1 inhibitor or siRNA (Figure 3B). Therefore, we examined the downregulation of Mcl-1 protein at the post-translational level using a protein synthesis inhibitor (cycloheximide (CHX)), and proteasome inhibitors (MG132 and lactacystin). Combined treatment with CHX and PTC596 rapidly degraded Mcl-1 expression compared to CHX alone (Figure 3C). In addition, PTC596-mediated Mcl-1 downregulation was reversed by MG132 and a lactacystin proteasome inhibitor (Figure 3D). These data indicated that PTC596 decreased Mcl-1 protein expression through the proteasome pathway. To prove the role of Mcl-1 downregulation in PTC596-induced apoptosis, we overexpressed Mcl-1 in Caki cells. Ectopic expression of Mcl-1 attenuated PTC596-induced apoptosis (Figure 3E). These results suggest that Mcl-1 downregulation is involved in PTC596-induced apoptosis.

### 2.4. Downregulation of DUB3 Is Associated with PTC596-Mediated Mcl-1 Degradation and Apoptosis

We investigated Mcl-1 ubiquitination by PTC596 and found that PTC596 induces ubiquitination of Mcl-1 protein (Figure 4A). The E3 ligases and deubiquitinases (DUBs) play an important role in the post-translational regulation of Mcl-1 protein through the ubiquitin-proteasome system [23]. To explore the relationship between Mcl-1 degradation via the ubiquitin-proteasome system (UPS), we examined the expression levels of various E3 ligases and DUBs for Mcl-1. While the expression levels of E3 ligases (FBW7 and β-TrCP) did not change, the expression levels of USP1 and DUB3 were decreased by PTC596 treatment (Figure 4B,C). When we investigated the BMI-1-mediated DUBs regulation, BMI-1 siRNA treatment markedly downregulated DUB3 protein level, but not USP1 (Figure 4D). In addition, knockdown of DUB3 also downregulated Mcl-1 protein levels (Figure 4E). Next, we examined the involvement of DUB3 in PTC596-mediated apoptosis using an overexpression system. Overexpression of DUB3 inhibited PTC596-induced apoptosis (Figure 4F). Taken together, these results indicate that DUB3 is involved in PTC596-induced Mcl-1 degradation and apoptosis.

### 2.5. The Apoptotic Effects of PTC596 in Various Cell Lines

We found that PTC596 did not alter DUB3 mRNA expression (Figure 5A). Therefore, these data suggest that PTC596-mediated DUB3 downregulation is modulated at the post-transcriptional level.

To prove the common anti-cancer effect of PTC596, we examined its effects on various cancer cell lines. As expected, PTC596 increased the sub-G1 population and PARP cleavage in human hepatocellular carcinoma (SK-Hep1), prostate carcinoma (DU145), lung carcinoma (A549), and cervical cancer (HeLa) cell lines (Figure 5B). Moreover, the expression levels of Mcl-1 and DUB3 were decreased by PTC596 in all tested cancer cell lines (Figure 5B). In contrast, PTC596 did not affect cell morphology or sub-G1 population in two normal cell lines (mesangial cells (MC) and kidney cells (TCMK-1)) (Figure 5C). Therefore, these results indicate that PTC596 induces apoptotic cell death in cancer cells, but not in normal cells.

## 3. Discussion

In this study, we found that pharmacological inhibition and knockdown of BMI-1 increases apoptosis in cancer cells, but not in normal cells, and induces downregulation of Mcl-1 protein expression via the ubiquitination pathway. Downregulation of DUB3 by PTC596 was involved in apoptosis and Mcl-1 degradation. Therefore, our results suggest that the inhibition of BMI-1 induces DUB3-dependent Mcl-1 degradation, resulting in the enhancement of cancer apoptosis.

PTC-209 was identified as the first small-molecule inhibitor by Kreso et al., and they also verified inhibition of the self-renewal ability of colorectal cancer-initiating cells [24]. PTC-209 impairs cell proliferation and apoptosis in multiple myeloma and head and neck carcinoma cells [16,17]. Moreover, Dey et al. investigated the increase in autophagy-mediated caspase-independent necroptotic cell death [25]. PTC596 is generated as a second small-molecule BMI-1 inhibitor and is cell-permeable. The IC50 value of PTC596 (68–340 nM) is lower than that of PTC-209 (500 nM), and PTC596 accelerates the degradation of BMI-1 protein at nanomole dosages [20]. When we tested PTC-209 and PTC596-induced apoptosis in cancer cells, PTC596 (10 nM) increased apoptosis, but not PTC-209 (10 nM) (Figure 2A). Furthermore, PTC-209 has limited potency and poor pharmacokinetic properties, and did not enter clinical trials, whereas PTC596 has a favorable safety profile and entered Phase I clinical trials [19,24]. Therefore, PTC596 is more effective than PTC-209 for cancer therapy. Recently, PTC596 was newly discovered as a microtubule polymerization inhibitor that directly binds to tubulin [26,27]. Eberle-Singh and coworkers found that PTC596 induced mitotic arrest and apoptosis in multiple pancreatic ductal adenocarcinoma cell lines. Moreover, high concentration of PTC596 (1~10 μM) decreased multiple tubulin genes in BMI-1-independent manner, resulting in the induction of mitotic arrest through inhibition of microtubule polymerization [26]. Since our results indicate that low concentrations of PTC596 (10 nM) increased cancer cell apoptosis through BMI-1-dependent Mcl-1 downregulation, we ruled out the effect of PTC596 as a novel small-molecule tubulin binding agent.

Nishida et al. reported that PTC596 downregulates Mcl-1 expression in a proteasome-dependent manner in acute myeloid leukemia [20]. However, they did not investigate the detailed mechanisms of Mcl-1 regulation by PTC596. We also found similar results from PTC596 and BMI-1 siRNA treatment (Figure 3A), and demonstrated that proteasome inhibitors (MG132 and lactacystin) prevented PTC596-induced Mcl-1 downregulation (Figure 3D). Protein ubiquitination via the regulation of UPS plays a crucial role in anti-apoptotic protein degradation, including Bcl-2, Bcl-xL, and Mcl-1, resulting in the induction of cancer cell death [28]. Because E3 ubiquitin ligases and DUBs are associated with protein stability through the UPS pathway, we investigated the alteration of these enzymes, which are capable of regulating Mcl-1 [29,30,31,32,33]. As shown in Figure 4B,C, Mcl-1-targeting E3 ligases (FBW7 and β-TrCP) were not altered by PTC596, whereas USP1 and DUB3 were decreased. Xu et al. showed that the accumulation of Mcl-1 is induced by the ectopic expression of USP1 [30]. Moreover, Wu et al. examined the interaction of DUB3 with Mcl-1, thereby inhibiting ubiquitination of Mcl-1 [32]. To confirm direct regulation of USP1 and DUB3 by BMI-1, we inhibited BMI-1 expression using siRNA. Knockdown of BMI-1 decreased DUB3 protein expression, but not USP1 (Figure 4D). As a result, we theorized that PTC596-induced USP1 downregulation may be an off-target effect. To verify the contribution of DUB3 in Mcl-1 stability and PTC596-induced apoptosis, we performed knockdown and overexpression of DUB3. Knockdown of DUB3 markedly downregulated Mcl-1 protein levels (Figure 4E). However, ectopic expression of DUB3 prevented PTC596-induced apoptosis and Mcl-1 downregulation (Figure 4F). Therefore, our data suggest that DUB3 plays a critical role in BMI-1 inhibition-induced Mcl-1 destabilization.

In a previous study, Maeda et al. reported that PTC596 induces Bax activation and downregulation of Bcl-2 and Bcl-xL expression [21]. However, we did not identify alterations in Bax, Bcl-2, and Bcl-xL protein levels (Figure 3A). Maeda et al. used a high concentration of PTC596 (>150 nM), over ~15 fold compared to the low concentration of PTC596 (10 nM) used in our study. Therefore, we ruled out the potential of Bax, Bcl-2, and Bcl-xL in PTC596-induced apoptosis.

Collectively, our study illustrates that inhibition of BMI-1 induces apoptosis through DUB3-mediated Mcl-1 degradation. Therefore, PTC596 may be a potential drug for cancer therapy in the future.

## 4. Materials and Methods

### 4.1. Cell Culture and Chemicals

Human renal cancer cell carcinoma (Caki), human cervical cancer cell carcinoma (HeLa), human lung cancer cell carcinoma (A549), human prostate cancer cell carcinoma (DU145), human hepatocellular cancer cell carcinoma (SK-Hep1), and kidney normal cells (TCMK-1) were obtained from the American Type Culture Collection (Manassas, VA, USA). Human mesangial cells were obtained from Lonza (Basel, Switzerland). Cell culture media were supplemented with 10% fetal bovine serum (Welgene, Gyeongsan, Korea), 1% penicillin–streptomycin, and 100 μg/mL gentamycin (Thermo Fisher Scientific, Waltham, MA, USA). PTC596 was purchased from Selleck Chemicals (Houston, TX, USA). Sigma Chemical Co. supplied cycloheximide, MG132, and anti-actin (St. Louis, MO, USA). z-VAD-fmk was obtained from R&D Systems (Minneapolis, MN, USA). Antibodies against PARP, anti-Bcl-xL, anti-Mcl-1, anti-BMI-1, and anti-cleaved caspase-3 antibodies were provided by Cell Signaling Technology (Beverly, MA, USA). Lactacystin and anti-caspase-3 antibodies were purchased from Enzo Life Sciences (San Diego, CA, USA). BD Biosciences (San Jose, CA, USA) offered anti-XIAP, anti-Bax, and anti-Bim antibodies. Anti-Mcl-1, anti-cIAP1, anti-cIAP2, anti-Bcl-2, and anti-β-TrCP were obtained from Santa Cruz Biotechnology (St. Louis, MO, USA). Abnova supplied anti-USP9X antibody (Taipei City, Taiwan). Anti-USP1 and anti-FBW7 antibodies were purchased from Bethyl Laboratories (Montgomery, TX, USA). Novus Biologicals supplied anti-DUB3 antibodies (Centennial, CO, USA). Anti-OTUD1 was purchased from Merck (Kenilworth, NJ, USA).

### 4.2. Cancer Cell Stem-Like Cell Population Analysis

To analyze the population of cancer stem-like cells, cancer cells were treated with PTC596 for 12 h. The ALDEFLUOR^TM^ Kit for ALDH assay (STEMCELLTM TECHNOLOGIES) (Cambridge, MA, USA) was used to measure stem cells, which were then analyzed by flow cytometry (BD Biosciences, San Jose, CA, USA).

### 4.3. Flow Cytometry and Western Blotting Analysis

To calculate the sub-G1 population, cells were mixed with 100% ethanol, incubated in 1.12% sodium citrate buffer containing RNase at 37 °C for 30 min, then added to 50 μg/mL propidium iodide, and finally analyzed using a Guava^®^ easyCyte™ flow cytometer (Luminex Corporation, Austin, TX, USA). To measure the fluorescence of the cancer stem cell marker CD44, harvested cells were suspended in PBS, incubated with APC-conjugated CD44 antibody (Invitrogen, Carlsbad, CA, USA) for 1 h at room temperature, and analyzed by flow cytometry. To investigate the alteration of protein expression, the total lysates were obtained in RIPA lysis buffer (20 mM HEPES and 0.5% Triton X-100, pH 7.6), then proteins were separated by SDS-PAGE and transferred to nitrocellulose membranes (GE Healthcare Life Science, Pittsburgh, PA, USA) [34]. Finally, protein expression was checked using an iBrightTM Imager (Thermo Fisher Scientific, Waltham, MA, USA).

### 4.4. Annexin V staining

To detect apoptotic cells, Annexin V was used to distinguish the cell death mode. The cells were washed twice in cold PBS and resuspended in Annexin V–binding buffer. This suspension (100 μL) was stained with 5 μL of FITC-conjugated Annexin V (BD Pharmingen, San Jose, CA, USA). The cells were incubated for 15 min at room temperature in the dark. The cells were analyzed using a flow cytometer (Accuri C6, BD Biosciences, San Diego, CA, USA).

### 4.5. DNA Fragmentation and DAPI Staining

To measure DNA fragmentation, a cell death detection ELISA plus kit (Boehringer Mannheim, Indianapolis, IN, USA) was used. Cell nuclei condensation was stained with 300 nM 4,6-diamidino-2-phenylindole solution (Roche, Mannheim, Germany) and then analyzed by fluorescence microscopy.

### 4.6. Asp-Glu-Val-Aspase Activity Assay

To detect caspase-3 activity, Caki cells were treated with PTC596 for 24 h, harvested, and incubated with reaction buffer containing substrate [acetyl-Asp-Glu-Val-Asp p-nitroanilide (Ac-DEVD-pNA)] [35].

### 4.7. Reverse Transcription-Polymerase Chain Reaction (RT-PCR) and Quantitative Polymerase Chain Reaction (qPCR)

RNA was isolated using TRIzol reagent (Life Technologies, Gaithersburg, MD, USA). cDNA was obtained using M-MLV reverse transcriptase (Gibco-BRL, Gaithersburg, MD, USA). Blend Taq DNA polymerase (Toyobo, Osaka, Japan) with primers was used to target genes: Mcl-1 (forward) 5′-GCG ACT GGC AAA GCT TGG CCT CAA-3′ and (reverse) 5′-GTT ACA GCT TGG ATC CCA ACT GCA-3′; USP1 (forward) 5′-TGA ACT TGC CAC TCA GC-3′ and (reverse) 5′-TTT CAC ACT TCC CCA AGT CC-3′; DUB3 (forward) 5′-CCG GGA GCA CTC TCA AAC AT-3′ and (reverse) 5′-TTG ACA CTC TGA GCT GCC TG-3′; and actin (forward) 5′-GGC ATC GTC ACC AAC TGG GAC-3′ and (reverse) 5′-CGA TTT CCC GCT CGG CCG TGG-3′. The amplified products were separated by electrophoresis on a 2% agarose gel and detected under ultraviolet light. For real-time PCR, SYBR Fast qPCR Mix (Takara Bio Inc., Shiga, Japan) was used, and the reactions were analyzed on a Thermal Cycler Dice^®^ Real Time System III (Takara Bio Inc., Shiga, Japan). The following primers were used to amplify the target genes: Mcl-1 (forward) 5′-ATG CTT CGG AAA CTG GAC AT-3′ and (reverse) 5′-TCC TGA TGC CAC CTT CTA GG-3′; USP1 (forward) 5′-TCT GTG CCT GCG TTG TTT GA-3′ and (reverse) 5′-CGT CCT TTG AAA TTG CCG GT-3′; DUB3 (forward) 5′-CCT CCC GAC GTA CTT GTG AT-3′ and (reverse) 5′-CAT GGA CTC CTG ATG TGT CG-3′; and actin (forward) 5′-CTA CAA TGA GCT GCG TGT G-3′ and (reverse) 5′-TGG GGT GTT GAA GGT CTC-3′.

### 4.8. Transfection

BMI-1 siRNA was purchased from Santa Cruz Biotechnology (St. Louis, MO, USA). USP1 siRNA, DUB3 siRNA, and control siRNA were purchased from Bioneer (Daejeon, Korea). For siRNA knockdown, Caki cells were transfected using Lipofectamine^®^ RNAiMAX Reagent (Invitrogen). DUB3 plasmids was transfected using LipofectamineTM 2000 (Invitrogen) in Caki cells.

### 4.9. Deubiquitination Assay

To detect the ubiquitination of Mcl-1, Caki cells were transfected with HA-ubiquitin plasmid and treated with MG132 (0.5 mM) and PTC596. The cells were harvested, washed with PBS containing 10 mM N-ethylmaleimide (NEM) (EMD Millipore, Darmstadt, Germany), resuspended in 100 mL PBS/NEM containing 1% SDS, and boiled for 10 min at 95 °C. RIPA lysis buffer with NEM (5 mM) was added to the lysates, which were dissolved using a l mL syringe four times and centrifuged at 13,000× *g* for 10 min at 4 °C. The supernatant was added to the primary antibody of the target protein overnight and then reacted with protein G agarose for 2 h. After centrifugation, the cells were washed twice with lysis buffer containing 1 mM PMSF and 5 mM NEM. The pellet was added to 2× sample buffer and incubated for 10 min at 95 °C. Ubiquitinated Mcl-1 was detected by Western blotting under denaturation conditions with HRP-conjugated anti-Ub (Enzo Life Sciences, San Diego, CA, USA).

### 4.10. Statistical Analysis

The data were analyzed by one-way ANOVA and post-hoc comparisons (Student–Newman–Keuls) using the Statistical Package for Social Sciences software (version 25.0; SPSS Inc., Chicago, IL, USA).

## 5. Conclusions

This study demonstrates that inhibition of BMI-1 decreases the cancer stem-like cell population and induces apoptotic cell death in cancer cells, but not in normal cells. PTC596 and BMI-1 siRNA induce Mcl-1 destabilization by downregulation of DUB3.

## Figures and Tables

**Figure 1 ijms-22-10107-f001:**
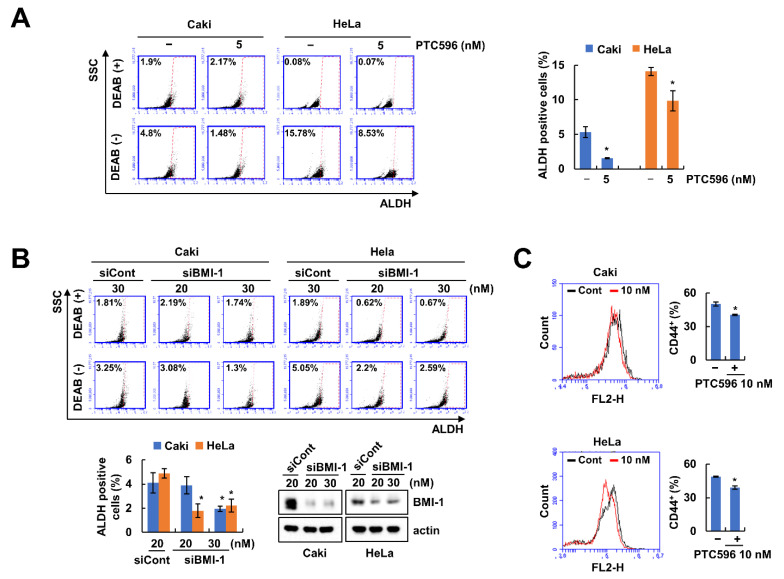
Inhibition of BMI-1 decreases the cancer cell stem-like cell population: (**A**) Caki and HeLa cells were treated with BMI-1 inhibitor PTC596 (5 nM) for 12 h; (**B**) Caki and HeLa cells were transfected with Control (Cont) or BMI-1 siRNA for 24 h. The population of stem-like cells was detected by flow cytometry and the protein expression was quantified by Western blotting (**B**); (**C**): The CD44 cancer stem cell marker was analyzed by flow cytometry in PTC596-treated Caki and HeLa cells. * *p* < 0.05 compared to the control.

**Figure 2 ijms-22-10107-f002:**
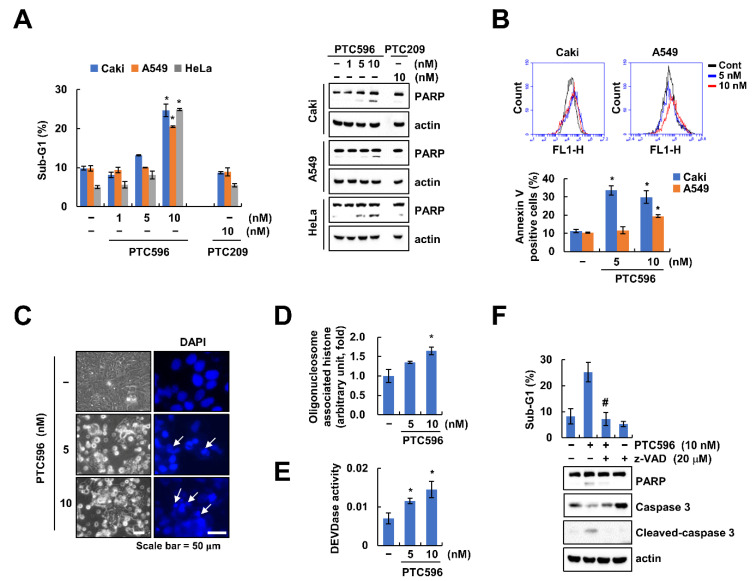
PTC596 induces apoptosis in a caspase-dependent manner. (**A**) Cancer cells (Caki, A549 and HeLa) were treated with PTC596 (1, 5, 10 nM) or PTC209 (10 nM) for 24 h; (**B**–**E**) Caki (**B**–**E**) and A549 (**B**) cells were treated with PTC596 (5, 10 nM) for 24 h. Cell death was determined by staining with Annexin V-FITC dye (**B**). Cell morphology and nuclear condensation were examined using a microscope (**C**). Fragmentation of the nuclei was examined using a DNA fragmentation assay kit (**D**). Caspase-3 activity was detected using DEVDase substrate (**E**); (**E**,**F**) Caki cells were pre-treated with z-VAD-fmk (z-VAD, 20 μM) and then treated with PTC596 (10 nM) for 24 h. The sub-G1 population and protein expression were measured by flow cytometry and Western blotting, respectively. The values in graphs (**A**,**B**,**D**–**F**) represent the mean ± SD of three independent samples. * *p* < 0.05 compared to the control. # *p* < 0.05 compared to the treatment of PTC596.

**Figure 3 ijms-22-10107-f003:**
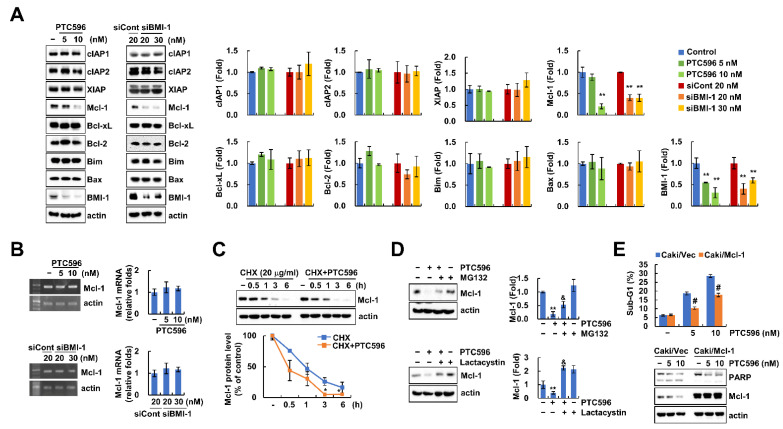
Downregulation of Mcl-1 is involved in PTC596-induced apoptosis. (**A**,**B**) Caki cells were treated with PTC596 (5, 10 nM) for 24 h or transfected with Cont and BMI-1 siRNA for 24 h; (**C**) Caki cells were treated with or without PTC596 (10 nM) in the presence of cycloheximide (20 μg/mL) for the indicated time periods; (**D**) Caki cells were pre-treated with MG132 (1 μM) and lactacystin (2.5 μM) for 30 min, then treated with PTC596 (10 nM) for 24 h; (**E**) Vector cells (Caki/Vec) and Mcl-1 overexpressed cells (Caki/Mcl-1) were treated with PTC596 (5, 10 nM) for 24 h. The sub-G1 population and protein expression were measured by flow cytometry and Western blotting (**A**,**C**–**E**). The mRNA levels of Mcl-1 and actin were quantified by RT-PCR (**B,** upper panel) and real-time PCR (**B**, lower panel). The band intensity was examined using Image J (**A**,**C**,**D**). The values in graphs (**A**–**E**) represent the mean ± SD of three independent samples. ** *p* < 0.01 compared to control. * *p* < 0.01 compared to CHX alone. ^&^ *p* < 0.05 compared to PTC596. # *p* < 0.05 compared to the PTC596 in Caki/Vec cells.

**Figure 4 ijms-22-10107-f004:**
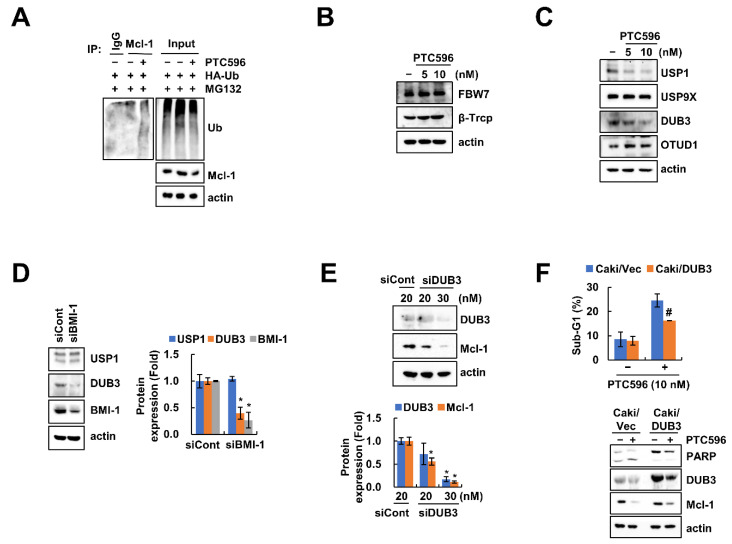
Downregulation of DUB3 is associated with PTC596-induced Mcl-1 downregulation and apoptosis. (**A**) Caki cells were transfected with HA-ubiqutin (HA-Ub) plasmid and treated with MG132 (0.5 μM) and PTC596 (10 nM). Mcl-1 ubiquitination was detected by Western blotting using HRP-conjugated anti-Ub antibody; (**B**,**C**) Caki cells were treated with PTC596 (5, 10 nM) for 24 h; (**D**–**E**) Caki cells were transfected with Cont, BMI-1 (**D**) or DUB3 siRNA (**E**) for 24 h; (**F**) Caki cells were transfected with Caki/Vec or Caki/DUB3 plasmids and then treated with PTC596 (10 nM) for 24 h. The sub-G1 population and protein expression were measured by flow cytometry (**F**) and Western blotting (**A**–**F**). The band intensity was examined using Image J (**D**,**E**). * *p* < 0.01 compared to the control siRNA. # *p* < 0.05 compared to the PTC596 in Caki/Vec cells.

**Figure 5 ijms-22-10107-f005:**
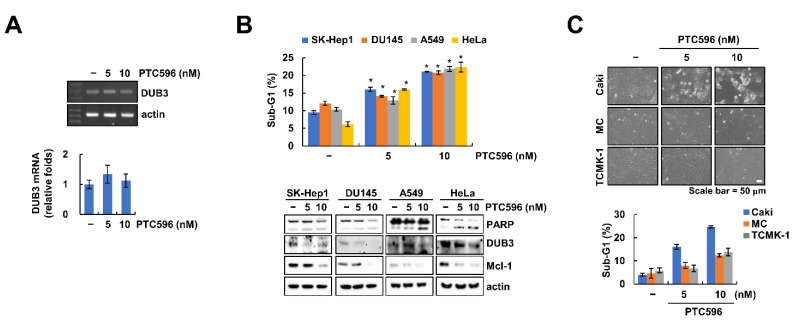
PTC596 induces apoptosis in various cancer cells. (**A**) Caki cells were treated with PTC596 (5, 10 nM) for 24 h. The mRNA levels were quantified by RT-PCR (upper panel) and real-time PCR (lower panel); (**B**,**C**) Cancer (SK-Hep1, DU145, A549, and HeLa) and normal cells (MC and TCMK-1) were treated with PTC596 (5, 10 nM) for 24 h. The sub-G1 population and protein expression were measured by flow cytometry and Western blotting (**B**,**C**). Cell morphology was measured by interference light microscope (**C**). * *p* < 0.01 compared to the control.

## Data Availability

The data to this study can be shared upon reasonable request from the corresponding author.

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
