# Peer review of "Inhibition of BMI-1 Induces Apoptosis through Downregulation of DUB3-Mediated Mcl-1 Stabilization"

_ijms, 2021, doi:10.3390/ijms221810107_

Round 1

Reviewer 1 Report

Authors demonstrated that BMI-1 inhibition induces MCL-1 destabilization by downregulation of USP1 and DUB3 deubiquitinase expression. However, article needs additional details to be more convincent and more appelable.

  1. Please, provide the inhibition levels of BMI-1 after PTC596 treatment, as WB or flow cytometry.
  2. figures 1a and 1b are not so representative, as weel as the CD44 expression. Please, provide figures with more impact, . May be, shoud be usufull to put a % in each dot plot or include more representative images.
  3. Please, provide protein quantification for WB analysis in order to provide an histogram bars with statistic.
  4. From figure 3a with the inhibitor seems also that BclXl and BIM are down-regulated. Is tere any explanation? Is this data consistent? I think that with a protein quantification for WB analysis all should be more clear. Why basal levels of the same protien (i.e are different from cells with inibitor and cells from siRNA treatments? Please, provide here leves for BMI-1 also for inibitor and not only for siRNA. Please, provide standard deviation for figure 3c. Please, provide a graph for figure 3d.
  5. From my point of view, from figure 4a is not so evident an increase of Ub. Usually, Ub changes are more evident.
  6. Could the authors provide data on the production of spheroids in the presence of MCL1 overexpression?
  7. Could the authors provide data aout annexinV positive cells after inhibitor or siRNA treatments?

Author Response

Could you find attached a file?

Thank you

Sincerely

Reviewer 2 Report

Many of the results presented in this manuscript are confirmatory. The effects of BMI-1 inhibition on cell death and Mcl-1 levels was already well known, as the Authors recognize. The novelty is thus limited to the involvement of USP1 and DUB3 in the process. This is however of some interest, and the paper may be publishable after the following points are addressed:

PTC596 is also a tubulin-binding drug. Can the Authors rule out a contribution of this (and/or other) mechanism(s) to their observations? This should be discussed.

The Authors mention (line 72, Fig. 2) that PTC209 was ineffective. This sounds strange since both PTC596 and PTC209 are considered to be BMI-1 inhibitors. Is it just a matter of affinity/concentrations used? This also needs to be mentioned (in Discussion, second paragraph, where PTC209 is discussed).

In Fig. 3A, left-side blots, there actually seems to be a decrease of Bcl-2, BclxL, and perhaps also Bim and Bax in the presence of 10 nM PCT596. The decrease, if any, is much less than that of Mcl-1, but the point needs attention in view of the results by Maeda et al. (ref 21), and a quantification ought to be shown to support the statements made on lines 191-196.

To make the case more watertight, the expts of fig. 4A-C need to be repeated with anti BMI-1 siRNA (in addition to the data with PTC596). If USP1 and DUB3 downregulation is downstream of BMI-1 inhibition, one would expect to find the same trend by decreasing BMI-1 expression. In the absence of such a “control” experiment the statement made in Conclusions seems to be an oversell, since there is no direct proof that BMI-1 siRNA downregulates the deubiquitinases.

In fig. 4B there actually seems to be a PTC596-related decrease of band intensity in the case of beta-trCP.

Fig. 5: it would be interesting to see a comparison of BMI-1 expression in the cancerous vs normal cell lines used.

As mentioned by the Authors, Nishida and coworkers found that Mcl-1 degradation in PTC596-treated cells was partly caspase-dependent. The Authors used z-VAD-fmk to verify that PCT596-induced death is by apoptosis. They may well verify whether the caspase inhibitor (by itself, without proteasome inhibitors) affects Mcl-1 levels. This also in view of the fact that reversal of Mcl-1 degradation by MG132 is partial (Fig. 3D).

A very recent paper published in Int J Mol Sci by Dong Eun Kim et al. reports that Mcl-1 post-transcriptional downregulation is induced by BMI-1026, a CDK inhibitor, in Caki cells. The study should be discussed and compared to the present one.

Minor/language

line 32: “can be major role”: can have a major role

line 39: “may an effective target”: may be an effective target

line 48: insert the word “inhibitors” after “receptor” (?; Mcl-1, MEK etc. are not inhibitors)

line 96: “using by protein synthesis inhibitor”: using protein synthesis inhibitor

line 103: “Ectopic of”: Ectopic expression of

line 113 and 154: Are “RT-PCR” and “real-time PCR” not the same thing?

line 142: “we also examined in various cancer cell lines”: we also examined its effects on various…

line 165 and 180: “resulted”: resulting

line 183: “targeted”: targeting (?)

Author Response

(The authors gave the same response as above.)

Round 2

Reviewer 1 Report

Authors responded to all requests by implementing the work with new experimental data. In this form the work can be considered for publication

Author Response

Thank you very much for your detailed and thoughtful comments.

Reviewer 2 Report

The manuscript has been improved, but unfortunately the results of the key control experiment I had suggested (point 4 in the first round of review) were not as expected. The Authors had concluded that BMI-1 was upstream of the deubiquitinases on the basis of the effect of PTC596, a BMI-1 inhibitor. They attributed the downregulation of USP1 and DUB3 to inhibition of BMI-1. However, it now turns out that silencing BMI-1 has no effect on USP1 and only “a slight effect” on DUB3. This latter result is shown only as an exemplary (the only?) WB, with no repetitions reported and no quantifications shown. The main message/finding of the paper was that Mcl-1 reduction was due to downregulation of the deubiqitinases downstream of inhibition of BMI-1. But this conclusion is now dramatically weakened: only one of the two deubiquitinases originally identified may be involved, and this now becomes the main message (that BMI-1 was involved in Mcl-1 downregulation was already known). And the evidence presented is insufficient. So, now there seems to be little in the paper that is novel and trustworthy. I ought to recommend rejection, but if the Editor agrees I am willing to give the manuscript another chance. The Authors must however fully and impeccably document that BMI-1 downregulation by siRNA leads to DUB3 (but not USP1, unless things change again) decrease.

Author Response

Thank you very much for your detailed and thoughtful comments.

Could you find attachment file?

Sincerely yours,

Round 3

Reviewer 2 Report

The Authors’ reaction to my comments has been exceptionally prompt. They evidently had already performed the experiments I requested. They now appropriately show these results and they have suitably modified the text of the manuscript, which can now be accepted for publication. One last comment: I believe the title should read “Inhibition of BMI-1 Induces Apoptosis Through Downregulation of DUB3-Mediated Mcl-1 Stabilization” (not “Destabilization”). Also, there are now a few language imperfections (for example, on line 17: was, not were). A final polishing before going online is advisable.